# Perception of COVID-19 Restrictions on Daily Life among Japanese Older Adults: A Qualitative Focus Group Study

**DOI:** 10.3390/healthcare8040450

**Published:** 2020-11-01

**Authors:** Risa Takashima, Ryuta Onishi, Kazuko Saeki, Michiyo Hirano

**Affiliations:** 1Department of Rehabilitation Science, Faculty of Health Sciences, Hokkaido University, Sapporo, Hokkaido 060-0812, Japan; risa-t@hs.hokudai.ac.jp; 2Department of Comprehensive Development Nursing, Faculty of Health Sciences, Hokkaido University, Sapporo, Hokkaido 060-0812, Japan; onishi.r1121@hs.hokudai.ac.jp; 3Faculty of Nursing, Toyama Prefectural University, Toyama 930-0975, Japan; saeki-k@pu-toyama.ac.jp; 4Hokkaido University, Sapporo, Hokkaido 060-0812, Japan

**Keywords:** COVID-19, community-dwelling older adults, urban, rural, inductive content analysis

## Abstract

The coronavirus disease 2019 (COVID-19) pandemic has exposed older adults to health and social risks. This study examined the perceptions of community-dwelling older adults regarding how COVID-19 restricted their daily lives. Six focus-group interviews were conducted with 24 participants (mean age, 78.2 ± 5.5 years) living in urban and rural areas in Japan. Then, a qualitative inductive content analysis was performed. Six themes were generated: “fear of infection and public, watchful eyes,” “consistency in daily personal life,” “pain from reducing my social life,” “readiness to endure a restricted life,” “awareness of positive changes in myself,” and “concern for a languishing society.” There was no change that would make their lives untenable, and they continued their daily personal lives at a minimum level. However, their social lives were reduced, which over the long term can lead to a lost sense of purpose in life. This was reported as an adverse factor in the development of other diseases and functional decline in previous studies. While there is no doubt that infection prevention is important, supporting older adults in engaging in activities that provide a sense of purpose in life could contribute to their present and future overall health including mental health.

## 1. Introduction

Although people of all ages are at risk of contracting coronavirus disease 2019 (COVID-19), previous studies showed that people older than 60 years are more likely to catch the disease [1], experience more severe symptoms [2,3], and have higher mortality rates [3,4,5]. In order to prevent the spread of COVID-19, many countries have adopted positive containment measures, such as a national lockdown, social distancing, and quarantine [6]. Although strict public health measures can protect older adults from infection, these measures can cause other problems, such as social isolation, limited access to routine health and social care services, and poor self-health management.

In particular, many experts have observed that as social distancing measures are implemented globally, social isolation and loneliness may increase among older adults [7,8,9,10,11]. Williams et al. [12] investigated the public perception and experience of social exclusion and isolation associated with the COVID-19 pandemic during the early stages of the United Kingdom’s lockdown. Focus group interviews were conducted with 27 British citizens over the age of 18. The qualitative analysis identified the following four main themes: participants’ sense of loss, government criticism, social distancing self-adherence, and uncertainty about social reintegration [12]. This study revealed the general public’s perceptions and actual experiences of social exclusion and isolation. However, this study did not include citizens over the age of 70.

However, several cross-sectional studies of social isolation and loneliness during the COVID-19 pandemic, including a sample of older adults, have been conducted. Tilburg et al. [13] measured social and emotional loneliness and the mental health of older adults (65–102 years old) in May 2020, two months after the implementation of a policy to keep citizens physically distanced, and compared their results to their earlier sample collected in the Netherlands in October and November 2019. The results showed a slight increase in social loneliness and a large increase in emotional loneliness, but mental health levels remained mostly stable. Meanwhile, Pierce et al. [14] examined the changes in mental health among British adults before and after lockdown. The results showed that the mental health of British adults had deteriorated compared with the trend before COVID-19, but it was particularly large in the younger age groups of 18–24 and 25–34 years. In addition, people who were employed before the pandemic worsened significantly. Steptoe and Fancourt [15] examined the predictors of loneliness during the COVID-19 pandemic using data from adults in the United Kingdom. They found that young adults, women, people with low education and income, people who were not economically active, people living alone, and people living in urban areas were at increased risk of loneliness. Luchetti et al. [16] used a nationwide sample of American adults and reported that older adults overall reported less loneliness compared with younger adults, although there was an increase in loneliness during the acute phase of the COVID-19 outbreak. Overall, these studies suggested that the problem of loneliness in older adults may not be as severe as in young adults during the COVID-19 pandemic.

Using data from 27 countries, including Asia, Pacific, Americas, Europe, and Middle East areas, Daoust [17] analyzed how older adults responded to the COVID-19 pandemic. Prospective self-isolation (if they were told to do so) increased with age and peaked at ages 70–75 but declined thereafter. Willingness to isolate increased gradually until age 70 but leveled off after that. Adherence degree with preventive measures for COVID-19 was not significantly affected by age and was essentially the same for all age groups. However, older adults were more positive about avoiding public transportation and avoiding small gatherings and visitors. In contrast, age had a negative effect on wearing a mask outside the home, with the 60–80-year-old age group showing a lower adherence to this precaution than the younger age group. This survey showed that the perception of being restricted or having to restrict behavior does not increase linearly with age. In some cases, people over 70 years old were less aware of restrictions. This perception among older adults may reduce the negative effects on social isolation and loneliness caused by social distancing and “stay-at-home“ measures. However, to date, there has been no investigation into how COVID-19 alters the actual daily lives of older adults and what distinct needs arise among them.

It is very important to know, from the viewpoint of older adults, how their lives are affected by COVID-19 to propose more appropriate measures to COVID-19 that respect the needs and dignity of older adults. Therefore, this study aims to explore the perceptions of older adults living in urban and rural areas in Japan regarding how their daily lives have been restricted by the spread of COVID-19.

### Spread of COVID-19 in Japan

On 30 January 2020, the World Health Organization (WHO) announced that the outbreak of COVID-19 in Wuhan, Hubei Province, People′s Republic of China, fell under “PHEIC: Public Health Emergency of International Concern” [18]. In Japan, the first person who tested positive for COVID-19 was confirmed on 16 January 2020 [19]. The number of new positive cases per day increased rapidly from late March 2020, and the Japanese government declared a state of emergency in five prefectures on April 6th [20]. The scope of application was expanded nationwide on April 16th, and it was lifted on May 25th [20]. Although lockdown was not conducted during this period, the government requested citizens to stay at home, except for necessary activities such as shopping, and requested temporary public facility closures such as schools, childcare facilities, and movie theaters [20]. The Japanese term typically translated means “request” but is understood to be taken to mean “demand” with a strong expectation that those asked will obey the directives; however, there were no legal penalties if these were not followed [21]. As of September 2020, there were two peaks in the number of new positive COVID-19 cases per day in Japan; the first was 720 new cases on April 11, and the second was 1605 new cases on August 7 [19].

This study was conducted in Hokkaido, which is a preference in northern Japan. This area had the highest number of cases in Japan until mid-March 2020 [19]. Prior to the nationwide state of emergency, the governors of Hokkaido declared their own “state of emergency” for the period from February 28 to March 19 [22]. 

In Japan, even after the lifting of the state of emergency, the government requested citizens of all ages to take the following measures: avoidance of the “3Cs” (closed spaces, crowded places, and close-contact settings); establishment of a “new lifestyle” to prevent the spread of the infection, including basic infection control measures such as social distancing, wearing masks, and washing hands; and avoidance of the “3Cs” indoors, especially in daily life, in the workplace, in restaurants, etc. [20].

## 2. Materials and Methods 

### 2.1. Study Design

The study used a qualitative focus group research design [23]. A focus group is a technique involving the use of in-depth group interviews in which participants are selected because they are a purposive, although not necessarily representative, sampling of a specific population, and the interviews are “focused” on a given topic [23]. One of the characteristics of a focus group interview is its group dynamics; therefore, the type and extent of data obtained through group social interactions is often deeper and richer than that obtained from a one-to-one interview [23]. In this study, we focused on people who are of older age (≥65 years of age) living in urban and rural areas in northern Japan whose daily lives were under conditions where measures including social distancing were requested by the government.

### 2.2. Participants and Recruitment

Using convenience sampling, the authors asked the members of two older adult community groups, with whom they were involved in other intervention studies, to participate in focus group interviews. One was an older group in a city (A) in an urban area of Hokkaido. The group, founded by the authors to conduct the intervention study, had been active once a month for 10 months. The other was a group of older adults in a rural town (B). The group had been active once a month for eight years, and the authors have been conducting an intervention study in the group for 13 months. In a focus group interview, it is important for the members to participate in the discussion comfortably with each other [23]. While using a group that is meeting each other for the first time may be desirable in some cases, Kitzinger advocates the use of pre-existing groups, as acquisitions could relate to each other′s comments and participants may be more willing to challenge one another [24]. A total of 24 members in both groups agreed to participate in the focus group interviews. The participants’ demographic profiles are shown in Table 1.

### 2.3. Data Collection

Six focus group interviews with three to five participants in each group were conducted from July to August 2020. As described above, Japan had its peak of new positive cases for COVID-19 per day during that time as of September 2020. Krueger and Casey recommend a group of six to eight people for a group interview [25]. However, in this study, we conducted interviews with fewer participants to ensure adequate distance between each other for infection control.

The first, second, and last authors served as interviewers in two groups each. The first and last authors, both women, hold PhDs, while the second author (a man) holds an MSc. All interviewers were university researchers who have practical experience with older adult healthcare and conducting qualitative research. Participants reported their name, age, gender, family structure, and subjective economic status on the demographic questionnaire. The subjective economic status was asked about in four stages from “*very concerned*” to “*not concerned at all*.” Interviews were conducted in a public community facility. We prepared the following three questions as an interview guide: (i) “Although this was the first time you experienced self-restraint in your daily life during the spread of COVID-19, what puzzled you? What were you feeling and thinking?”; (ii) “How has your life changed due to the spread of COVID-19?”; and (iii) “Since the state of emergency was lifted at the end of May, has your daily life or have you personally changed over time?” However, the interview guide was used as a flexible tool, and participants’ free speech was respected. All interviews were conducted in Japanese (the researchers’ and participants’ native language), recorded, and transcribed verbatim. Each interview lasted 60 to 90 min. One participant was absent from the focus group interviews. In response to his request, he responded to the same questions by fax. Although the transcripts were not returned to participants, results, including relevant narrative descriptions, were shared with them, and they provided feedback. Excerpts from the interviews that appear in this paper were translated to English from the original Japanese transcripts.

This study followed the guidelines set out by the 1975 Declaration of Helsinki (2008 revision). It was approved as part of the intervention study described above by the Ethics Review Committee of Hokkaido University Faculty of Health Sciences (no. 18–39). Participants were given verbal and written explanations on the assurance of anonymity, the confidentiality of data, the assurance of free participation in research, and the publication of the results. All participants provided written informed consent for research cooperation. They were free to withdraw at any time. After the interviews, participants were given bottled green tea as a token of gratitude.

### 2.4. Data Analysis

A qualitative and inductive data analysis based on content analysis was performed [26]. The analysis process consisted of three main phases: (1) preparation, (2) organizing, and (3) reporting. In phase 1, the first author repeatedly read the data, identified the measuring units (parts of text) in the process, and coded the main themes. In phase 2, the codes were organized and categorized, and subthemes and themes were generated. Preliminary analysis results were shared with the second and last authors, and the organized data were modified until consent was obtained. As a result, six main themes composed of multiple subcategories were generated. The data became saturated when no new themes emerged. After consensus was reached among all authors, subthemes and themes were established. All analyses were done manually, in Japanese, and translated into English at the time of writing.

### 2.5. Study Rigor and Trustworthiness

The following strategies were used to increase the trustworthiness and credibility of data analysis [27,28]. The researchers performed a continuous comparative analysis and described the results of the analysis process in detail. In the group activities in which the participants gathered, a summary of the results of this study was shared, and the member check was carried out. After listening to the authors explain the results, participants discussed the results in groups for about 20 min and provided feedback to the authors. Participants generally agreed with the results and spoke more about what they felt about their lives during the spread of COVID-19. The methods and results followed the Consolidated Criteria for Reporting Qualitative Research guidelines—a 32-item comprehensive checklist used in reporting data from interviews and focus groups [29].

## 3. Results

Focus-group interviews were conducted with 23 older adults living in urban and rural areas of northern Japan to explore their perceptions on how COVID-19′s spread restricted their daily lives. In addition, one participant who was absent from the focus-group interview answered the same questions by fax. As a result of qualitative inductive content analysis, 373 codes were extracted and six themes were generated: “fear of infection and public, watchful eyes,” “consistency in my daily personal life,” “pain from reducing my social life,” “readiness to endure this restricted life,” “awareness of positive changes in myself,” and “concern for a languishing society.” The structure of the themes is shown in Table 2. Hereinafter, each theme is described in order. 

### 3.1. Fear of Infection and Public, Watchful Eyes

Participants expressed two types of fear. One fear was that they might have been infected with COVID-19. They talked about how this fear was heightened by the perception that being older and having a pre-existing condition were risks of more severe symptoms with COVID-19:


*“I′m afraid to go out and walk, after all.”*



*“Old people would die if they catch it (COVID-19). I thought I should be careful about this.”*



*“There is no safe place. I cannot go to even a convenience store or a market.”*


The other was the fear of public criticism. Characteristically, this fear was extracted only from the narratives of rural participants. For example, participants said that if they were to walk into a store or attend a gathering without wearing a mask, they would be looked at coldly whether they were young or old. In addition, in determining whether or not an activity could be carried out, the focus was on how to judge the surroundings rather than whether there was a risk of infection:


*“If I don′t wear a mask, I′ll be looked at coldly.”*



*“The kitchen [of a public community facility] still can’t be used. We can’t use it. You can use it, but if you say, ‘I used it,’ you will be criticized and asked, ‘What were you thinking?’ That′s why we cannot use it.”*


These two types of fears prompted participants to recognize the “readiness to endure this restricted life” described below and encouraged adherence to infection-prevention behaviors. Some participants described careful information gathering as a way to protect themselves and others around them from infection and public watchful eyes:


*“Coronavirus is always on my mind. So, I am listening here and there and watching the news.”*



*“The older you get, the more you believe the impossible stories that don′t have roots anymore. I have to be careful.”*


### 3.2. Consistency in My Daily Personal Life

Participants said that while they have lost their social connections and leisure activities, as described in theme of “pain from reducing my social life,” there has been little change in their daily personal lives:


*“My life is just about the same. Nothing special, um. I do not usually go to the neighborhood.”*



*“I don′t think my daily life has changed. No change, nothing special. It’s just that events were canceled.”*


The need to practice thorough hygiene to prevent infection, such as wearing masks and disinfecting hands, arose. However, it was recognized that these hygienic actions had been experienced before the spread of COVID-19, such as during the influenza season, and that they would only increase the degree of implementation. Therefore, participants said that there was little resistance and that it became a habit relatively easily:


*“Gargle, wash hands, disinfect, these are already my habit. Everyone does them normally. You get out and come back, and then, you wash your hands and gargle and disinfect your hands with alcohol, do you not? After they are done, I start to act inside my house.”*


As a background, some participants mentioned that they were pensioners and hence, they were unlikely to feel the serious changes that would make their lives untenable. All persons aged 20 to 59 years who have an address in Japan participate in the National Pension, a public pension system, which provides benefits called “Basic Pension” due to old age, disability, or death [30]. In addition, employees also usually enroll in “Employees’ Pension Insurance systems.” Pensioners can receive Old-age Pension each year from age 65, depending on the period of enrollment and the amount of earned wage [30]. The average amount of pension generally guarantees the lowest standard of living in Japan. Two participants had paid jobs, but none of them were economically disadvantaged by reducing their activities. Therefore, it was said that there were no real changes in daily personal life.

### 3.3. Pain from Reducing My Social Life

While they did not feel much change in their daily personal lives, they realized that their social life had shrunk. They understood social restrictions, but spoke of pain, regret, and loneliness. They were convinced of the measure to reduce their social lives, but they talked about their pain, regret, and loneliness. Emotional loneliness was only reported by participants who lived alone. As the government instructed that people refrain from unnecessary and non-urgent activities, their daily lives had come to mainly consist of the minimum necessary activities. In particular, participants talked about how the number of activities to connect with others decreased. Every social activity was suspended or postponed. Family visits were also canceled or postponed. Interactions with family members who were hospitalized or admitted to the facility were also to be avoided. Participants who lived alone said they felt increasingly lonely after losing their connections with people. They said that the estrangement of their neighbors made them feel lonely and more anxious about living alone:


*“Everyone is becoming isolated gradually. Everyone is becoming solo, it’s lonely.”*



*“We were all depressed because we couldn′t go to the hot spring. We used to go there once a month… because most of us live alone.”*


Activities for leisure as well as to connect with people were typically interrupted. Since these activities were considered unnecessary and not urgent, participants felt that they should not engage in them owing to infection risk. Participants understood activity suspension, but also stated, “I’m feeling depressed” and “I’m under a lot of stress.”


*“I used to have a drinking party or lunch with young people from my retired workplace once a month, but now I can’t have any.”*



*“The Bon Festival dance will be also canceled this year. It′s just a festival. Unlike calling it off because of rain, in everyone is heartbroken, I think everyone has a strong sense of regret.”*


Bon Festival dance is a traditional Japanese dance performed every year in mid-August to welcome one′s ancestors.

Some participants expressed their confusion over the forced breakdown of their schedules. For many participants, activities were decreased. Some participants noted that it created life imbalances that they could not handle at the time:


*“I suddenly lost all schedules, and when I was told not to go out, not to walk, and to stay still at home, it devastated me considerably.”*


In contrast, participants who had to take care of their grandchildren, owing to school closures, talked about how they had become rapidly busier. One participant said, “I′ve lost my time,” and that the activities he wanted to engage in were no longer available even if he wanted to do them.

### 3.4. Readiness to Endure This Restricted Life

Although participants talked about “pain of reducing my social life,” they were prepared to endure the current limited life to the end, even if it would be prolonged for a long time. As noted above, the two types of fears strengthened participants’ readiness to continue to tolerate infection prevention for others and themselves:


*“You either die of the disease or stay at home and put up with it. It is whether you can bear the unbearable and endure the unendurable.”*



*“It′s not just for me. If you do not follow the rules, other people will be annoyed.”*


While participants looked forward to the day when the vaccine would end their restricted lives, they thought it would be a long time before that day came. They said they were prepared to continue their present “meager” daily life, even in a long battle, until things would be over:


*“This is probably not going to end in two or three years. Well, I feel like it′s going to be a long game.”*



*“Until a vaccine is developed, I’ll make a meager living. I have to be careful not to get infected.”*


To overcome the life crisis caused by COVID-19, participants talked about the three kinds of efforts they made in their daily lives. First, participants chose to engage in activities that they could do with the restrictions:


*“I′m still trying to limit my opportunities to go out, so at most I′ll go walking when there aren′t many people around. I try to avoid going anywhere and eating out for a bit, or shopping as much as possible.”*


In particular, there were frequent transitions to alternative activities with a low risk of infection or personal activities that could serve the same purpose. The participant, who was the organizer of the local summer festival, switched to radio calisthenics because of the risk of infection of the festival. As leisure activities declined, participants started playing games on their smartphones. Participants who were no longer able to join group sports or social activities said that they started walking or home gardening alone. Participants who originally focused on personal activities and activities with family members living together said they were not affected much because they were able to continue their activities without modification.

Second, frequent changes were made in activity styles and prioritized infection prevention. For example, regarding shopping, participants reduced the number of shopping trips, selected a time when customers were fewer, reduced the number of family members who went shopping, and shortened the time required for shopping by listing items to buy in advance. However, in some cases, participants mentioned changes in performance styles would undermine the value of their activities. One participant said that the change from face-to-face meetings to document circulation made it lose the meaning of discussing the agenda, replacing it with only seeking formal approval from the members. Several participants in both urban and rural areas were puzzled by changes in funeral practices. To reduce the risk of infection, only relatives were allowed to enter the ceremonial hall. Other people could no longer listen to sutras and pray at the altar to remember the deceased. In response to the change in the style of the traditional ritual that people are used to, one participant said the following:


*“At the funeral, I wanted to go to the altar, but I was sent back saying that only relatives are allowed to enter. That′s out of order.”*


Third, intentional or consequent reduction in the range of activities was mentioned. Due to the government′s recommendation to refrain from cross-regional travel as well as the suspension of various social activities, participants’ spheres of activity were limited to their houses and the surrounding areas. Avoiding the use of public transportation also reduced their spheres of actions:


*“I am told to restrain myself, so I have nowhere to go. So, I think I just have to move in my territory.”*



*“I have stopped going out to the city center, actually, I have stopped using public transportation.”*


Even if they were using their own cars, participants who live in the rural area said that they still refrained from going to other areas because of the “fear of public, watchful eyes” because their license plates would let people know that they came from other areas.

### 3.5. Awareness of Positive Changes in Myself

Some participants experienced positive changes in response to being forced to change their lifestyle. They said that fewer trips to bars or more walking alone made them feel healthier. Other participants noticed the convenience of using smartphones to interact with family members.

In addition, one participant said he became more aware that time and opportunity were finite, as COVID-19 prevented them from doing what they could have taken for granted. He said that uncertainty about the future made him feel that he would not miss the opportunity to participate actively and try his best every day:


*“If you have a chance, do it when you can, or don′t miss a lot of opportunities... Value the present. Nobody knows what is ahead. From here, the coronavirus may spread more, and the government may declare a state of emergency again.”*


### 3.6. Concern for a Languishing Society

This theme was generated only from male participants’ narratives. Participants were concerned not only about themselves, but also about the sense of stagnation in society as a whole. Some participants expressed frustration at the tendency for everything to be judged as “it can’t be done” and to be given up on before any discussion of how to do it:


*“Everyone says ’we can′t do this,’ and ‘we can′t do that’ without hesitation... if we don′t do what we can do first, we won′t be able to do it again... We need more power to lead, including the system, like pulling.”*



*“In conclusion, it was decided not to do it, but the way of proceeding of the executive committee [of a summer festival] was strange. Why did they decide from the beginning that they shouldn′t play the children’s portable shrine and ‘odoriyama’ (a traditional dance in the area)? They should have discussed whether they could do those programs or not before they made a decision.”*


Some participants expressed concerns about the future of the entire nation as society languishes due to their prolonged restrictions under COVID-19. One participant spoke of his concern that the closure of the school for several months had caused a significant delay in the education of his children. This participant compared young people living today with his younger self who was able to feel that his efforts had paid off and that he was helping the society. He was concerned that young people living in uncertainty about what the world would be like with COVID-19 might not know what to hope for:


*“I wonder, from now on, what young people or people around 20 years old will strive for with hope, or if they will have a future, though this may be an extreme idea.”*


## 4. Discussion

This study aims to explore the perceptions of older adults living in urban and rural areas in Japan regarding how their daily lives have been restricted by the spread of COVID-19. As a result of qualitative inductive content analysis, six themes were generated. Participants mentioned that there was little change in their daily personal lives. Although there were restrictions on pleasurable activities, it did not make their daily lives impossible. Therefore, it could be argued that it was characteristic of this older group of participants that they felt there was little change in terms of their minimal personal lives to get through each day. This is in agreement with the findings of Pierce et al. [14], who examined mental health changes in adults aged 16 and older, including those aged 70 and older, before and after lockdown in the United Kingdom. Mental health deterioration was particularly severe in younger age groups of 18–24 and 25–34 years and in people employed before the COVID-19 pandemic. In addition, Son et al. [31] investigated the impact of the COVID-19 pandemic on college students’ mental health in the United States. They reported that 71% of college students noted increased stress and associated anxiety, and 82% reported increased concerns about academic performance. In contrast, Tilburg et al. [13] investigated the social and emotional loneliness and mental health of older adults (65–102 years old) in the Netherlands, where the policy of maintaining physical distance had been implemented. They reported that their mental health was almost stable, although their loneliness increased. Participants in our study were restricted in their behavior, including cessation of activities, as noted in the theme of “pain from reducing my social life.” However, as the theme of “constancy of my daily personal life”’ indicated, their daily personal life seemed to be well maintained. Some participants pointed out the economic stability of their lives by receiving pensions. Therefore, the impact on mental health related to economic security/anxiety might be mitigated compared with the younger generation or those in employment.

Many participants talked about social isolation and that they found it difficult to connect with others in the theme of “pain from reducing my social life.” However, emotional isolation, such as loneliness and the apprehension of social isolation, emerged only from participants living alone. Bu, Steptoe, and Fancourt [15] compared sociodemographic predictors of loneliness before and during the COVID-19 pandemic using cross-cohort analyses of data on adults in the United Kingdom captured before and during the pandemic. They found that people living alone had a higher risk of being lonely. Furthermore, people who were already at risk of being lonely, including adults living alone, experienced a heightened risk during the COVID-19 pandemic compared to before. More specifically, living alone is an indicator of social isolation [32], and previous studies have repeatedly pointed out that living alone is a risk factor for loneliness in older adults [33,34]. In this context, older adults living alone may be more likely to experience loneliness under the COVID-19 pandemic. In particular, as older adults generally use information and communication technologies less than younger people, the loss of face-to-face interaction opportunities likely mean the loss of human connections. It is inferred that older adults living alone need more support than usual to prevent them from becoming lonely during the COVID-19 pandemic.

Tilburg et al. [13] responded to the small increase in social loneliness, but the large increase in emotional loneliness stating, “This might indicate that it was not so much the older adults′ social embedding that was affected by the crisis, but rather the ‘emptiness’ and close connectivity with people around them.” Through the expression “emptiness,” Tilburg et al. [13] points to a lost sense of meaning and purpose embodied in an activity rather than a loss of opportunity to engage in it. Participants in our study described “pain from reducing my social life.” In this theme, they talked about the loss of enjoyable activities as well as difficulties in maintaining connections with others and reduction in their spheres of action. In other words, they may have experienced a change in their daily lives in which meaningful, if unnecessary and non-urgent, activities were scrapped from their lives, and their daily lives were composed only of the minimum activities described in the “constancy of my daily personal life.” This change may be related to the sense of “emptiness” as mentioned by Tilburg et al. [13]. Moreover, participants showed an understanding of the high frequency of abandonment of social and leisure activities that were less necessary or urgent. As expressed in the theme of “readiness to endure this restricted life,” they were prepared to continue living with such restrictions, even for a long time. However, if this situation continues over the long term, it may damage participants’ health. To prevent nursing care, dementia, falls, depression, etc., and to maintain and improve older adults’ health, it is important for older adults to go out, interact with others, exercise, and participate in society. It has been shown that restricting these opportunities increased the risk of requiring nursing care, dementia, and early death as well as the severity of the condition requiring nursing care [35].

In addition, reduced social life can lead to a lost sense of meaning and purpose in life and “*ikigai”* for participants. *Ikigai* is the source of value in one′s life and is valuable on a personal level [36]. As noted above, social and leisure activities can be significant even if not necessary or urgent for older adults. In the focus-group study with British adults conducted by Williams et al. [12], one of the main themes, “loss,” included psychological and emotional “loss” such as loss of motivation, meaning, and self-worth. Previous studies reported that older adults with a higher sense of meaning and purpose in life are less likely to have adverse effects on health, including mortality, declines in physical function, fatigue, disabilities, Alzheimer′s disease, and clinical strokes [37]. Further, a stronger degree of *ikigai* is significantly associated with a lower risk of incident functional disability [38]. In contrast, a lack of *ikigai* is associated with higher mortality from strokes and/or cardiovascular diseases [39]. Thus, a reduced social life can deprive older adults of the sense of meaning and purpose in life, and *ikigai* and may cause a great loss to their health in the long run. Taking into account the long-term health effects of older adults, measures to deal with the risk of infection are a prerequisite; however, it is also necessary to support their participation in activities to connect with others and engage in leisure activities.

Finally, participants in this study included both urban and rural residents. In the theme of “fear of infection and public, watchful eyes,” the participants expressed two kinds of fears. The fear of “public, watchful eyes” was extracted only from the narratives of rural participants, thus reflecting a geographic divide. A sense of fear facilitated participants’ adherence to infection-control behaviors, as shown in the theme of “readiness to endure this restricted life.” In addition to the fear of COVID-19 infection, rural participants were also afraid of being criticized by others and tended to be more sensitive to the prevention of COVID-19 than participants in urban areas.

In this context, it is worth noting that Sato et al. [40] observed that Americans were more likely to trust strangers than Japanese people because American society has greater social mobility and higher social uncertainty than Japanese society does. Social uncertainty is a situation whereby people do not have enough information to predict others’ actions [41]. In addition, rural areas are considered to have less social mobility and less social uncertainty than urban areas in Japan [40]. This is because Japanese rural areas rural societies still prefer homogeneity and have closed communities [42,43]. Due to these sociocultural characteristics of rural areas, it is assumed that “public, watchful eyes” worked strongly among those participants. In countries and regions where there was a higher fear of “public, watchful eyes,” in addition to infection fears, people may be empowered to adhere more strongly to infection-prevention practices. This has a positive effect from the viewpoint of infection prevention; however, older adults may be more likely to lose their sense of meaning and purpose in life and *ikigai* as a result of reduced social lives.

### Limitations

This study conducted focus-group interviews to explore the perceptions of community-dwelling older adults about their daily lives restricted by the COVID-19 pandemic. The results of this study may help understand older adults’ perceptions in countries and regions where policies such as social distancing and staying at home limited daily life. However, considering the sociocultural impact, it should be taken into account that these results were the experience of older adults in Japan, where social mobility and uncertainty are lower than in Western countries. In Japan, the fear of “public, watchful eyes” had a stronger influence among participants living in a rural area. This result helps clarify the experience of older people living in communities that prefer homogeneity and form closed communities.

In the theme of “readiness to endure this restricted life,” participants said that they were prepared to continue to adhere to public health measures even in the long run; however, the impact of the social desirability bias needs to be carefully considered. Social desirability is the pressure for someone to claim to have followed a norm [44]. In this context, Daoust et al. [44] noted the impact of the social desirability bias on the investigation of compliance with COVID-19 public health measures and suggested that people adopt “face-saving” strategies based on their findings. Meanwhile, we conducted focus-group interviews using existing groups. Participants were familiar with each other and relatively comfortable discussing issues related to the pandemic and public health measures in groups. It might have mitigated the impact of the social desirability bias. However, it is possible that the “public, watchful eyes” also worked in this manner during focus-group interviews among rural participants. In other words, the rural participants might have been more resistant toward talking about non-compliant behavior because they were worrying about the “eyes” or opinions of other participants. Thus, the influence of social desirability bias should be considered to a greater extent when interpreting rural participants’ narratives.

The interviews were conducted 1.5–2.5 months after the lifting of the state of emergency in Japan. Different results may have occurred if the interviews were conducted during the state of emergency as restrictions on the lives of older adults were greater during the declaration. However, the state of emergency requested self-restraint in group activities, and interviews were not realistic given the characteristics of older adults who generally preferred face-to-face interactions over those online. In this study, we also asked about living under the state of emergency, and we believe that 1.5–2.5 months after the lifting of the state of emergency was close enough to get answers that adequately detailed the fresh experiences of older adults.

Of the 24 participants in the study, only eight (33%) were women. In the results of this study, the theme of “concern for a languishing society” was generated only from the narratives of male participants, which shows a gender difference. Further investigation may be necessary from the perspective of gender differences given the relevance of gender in health and well-being.

All participants in this study belonged to a community-based group for older adults who habitually participated in group activities. Under the state of emergency, group activities were suspended; however, they had resumed at the time the interviews were conducted. For this reason, these participants may have characteristics that make them less likely to feel social and emotional isolation. Even among the participants in this study, those living alone complained of emotional isolation. Thus, the impact of the COVID-19 pandemic on the lives of older adults who were not in the habit of participating in social activities needs further investigation.

## 5. Conclusions

The following six themes were generated for participants’ perceptions of their daily lives under the COVID-19 pandemic: “fear of infection and public, watchful eyes,” “consistency in my daily personal life,” “pain from reducing my social life,” “readiness to endure this restricted life,” “awareness of a positive change in myself,” and “concern for a languishing society.” A participant characteristic was that they felt less change in their daily personal lives because their lives would not flounder even if various activities were restricted. As previous studies [7,8,9,10,11] have repeatedly raised concerns, participants experienced social isolation caused by reduced social lives; however, the emotional loneliness and apprehension due to social isolation were mentioned only by participants living alone. Although support for the isolation and loneliness of people living alone has been emphasized in the past, further care for this group may be required during the COVID-19 pandemic. 

Even under limiting circumstances, many participants continued to engage in activities by choosing activities that considered the risk of infection or by changing the way they were performed. However, social activities that were often judged to be unnecessary and non-urgent were cut off, and individual activities became the focus. Characteristically, their spheres of actions were reduced, and activities to connect with people and leisure activities disappeared. This characteristic was more pronounced among rural participants who were afraid of not only infection but also public, watchful eyes. Participants showed an understanding of their inability to perform these activities and had a “readiness to endure this restricted life.” However, if this condition continues for a long time, it is feared that the sense of meaning and life purpose may decrease or their *ikigai* may be lost. These declines and losses are known to have adverse effects on the development of other diseases and functional decline and may compromise the health of older adults in the long run. While infection prevention is of course important in measures for COVID-19 among older adults, the long-term effects on their health should also be considered. For this purpose, healthcare professionals are expected to contribute to supporting activities for continuing their sense of meaning and purpose in life and *ikigai*.

## Figures and Tables

**Table 1 healthcare-08-00450-t001:** Participants’ demographic profile.

Demographic Items	*N* = 24
Gender	
Man	16 (66.7)
Woman	8 (33.3)
Age (years)	78.2 (5.5)
65–74	6 (25.0)
75–84	16 (66.7)
85+	2 (8.3)
Family composition	
Living alone	9 (37.5)
Living with spouse	10 (41.7)
Living with spouse and other family member(s)	5 (20.8)
Residential area	
Urban	10 (41.7)
Rural	14 (58.3)
Subjective economic status	
Very concerned	0 (0)
Somewhat concerned	15 (62.5)
Slightly concerned	8 (33.3)
Not concerned at all	1 (4.2)

Note: Data are presented as percentages (%) or mean (*SD*).

**Table 2 healthcare-08-00450-t002:** Theme structure.

Themes	Subthemes
Fear of infection and public, watchful eyes	Fear of being infected
	Fear of public watchful eyes
	Defense through vigilant intelligence gathering
Consistency in my daily personal life	Unchanging daily personal lives
	Good acceptance and habituation of daily hygiene behavior
Pain from reducing my social life	Difficulty of maintaining connections with people
	Loss of activities for pleasure in life
	Tightness that gradually built up
	Confusion due to the collapse of my schedule
Readiness to endure this restricted life	Patience with others and myself
	Readiness to overcome the prolonged war
	The long-awaited day of the end
	Selection of activities considering the risk of infection
	Changing the style of performance in which prevention of infection is prioritized
	Intentional/consequential reduction in my sphere of action
Awareness of positive changes of myself	Awareness of the finite nature of opportunity
	Expectations for a richer and healthier lifestyle
Concern for a languishing society	Resistance to the tendency to give up activities from the beginning
	Concern about the future of people′s lives

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
