# Peer review of "Perception of COVID-19 Restrictions on Daily Life among Japanese Older Adults: A Qualitative Focus Group Study"

_healthcare, 2020, doi:10.3390/healthcare8040450_

Round 1
Reviewer 1 Report
I read this study with a great deal of interest. I have already reviewed about a dozen research on COVID-19 and this is probably the most needed one. The topic is, needless to say, very important and timely, the research design is clean and complements other quantitative works, and makes a neat contribution to the literature on COVID-19 research. My verdict is as acceptance after minor revisions, which should be quite easy to implement. I am available to review the new version if needed. Below, I outline my comments.
-Abstract, last sentence: it seems that the "long-term effects on health" most specifically refer to mental health. It might be appropriate to be clearer by specifying it.
-Page 3: description of the case (Japan). In the last paragraph of section 1.1., I would be interested to know whether the government targeted, like many countries, a specific group of age. For example, some governments target the 65+ years old, other focus on the 70+, etc. Was there a recurrent group mentioned in the government's communication/ads?
-Table 1 seems to not fit the description of the note. It seems that for every line, the raw N is presented with the % in parentheses. No SD. Also on Table 1: the answer choices for subjective economic status are very odd. the option "Somewhat concerned" is lacking and there is no obvious difference between "a little concerned" and "not so concerned." Is this a translation issue? It would have been problematic for quantitative purpose but it's not a major issue in that case.
-Page 6: what do you mean by pensioners? A reader that is not familiar with Japan (like me) would not be sure to understand. Is this similar to long-term health care houses like in the US?
-I particularly liked the discussion about the male-only comments on page 8 and the relationship between loneliness and living with a parter or not. The distinction between rural/urban was also noteworthy.
-Page 9, the very end of the first paragraph of section 4. It should be clarified that the impact on mental health (that is mitigated) related to economic security/anxiety. It leaves intact other forms of mental health issues like loneliness (i.e. some can have no economic anxiety but still have mental health issues).
-About eh fear of "public, watchful eyes": did they talk about the fear of being judged by their partner? Of course it applies to only those who live with a partner, but I can see a potential fear of being seen as reckless if one 'goes out' too often. The fear would not come from the public, but rather within the household.
-The only more important comment that I have is about social desirability bias. The authors do not tackle this issue at all in their discussion about the research design. There are two things. First, I think it should be important to acknowledge that there is a social desirability bias when it comes to mention what kind of activities we have done. See Daoust et al. (2020; in JEPS) for a case study on Canada and/or Daoust et al. (2020; SSRN) for a replication in 12 countries. After all, the authors discuss the potential fear from the public, thus recognizing a social norm. Second, the implications should be discussed. In their case, the fact that the people who were part of the focus group were quite familiar with this exercice, the participants (and maybe the researchers?) might help in the sense that they might feel comfortable to mention freely that they sometimes go out more than they 'should'.
Again, I believe that this paper is much needed and very important. This is a masterpiece that demonstrates the importance of social sciences in our response to COVID-19. While my comments below would improve the manuscript, I think that they are quite easy to implement (hence the 'minor revision' verdict).
Author Response
Responses and revisions regarding the suggestions from reviewer: 1
|
No. |
Suggestions |
Responses and revisions |
|
1 |
-Abstract, last sentence: it seems that the “long-term effects on health” most specifically refer to mental health. It might be appropriate to be clearer by specifying it.
|
As you suggested, I agree that mental health problems such as loneliness are significant in the short term. However, as mentioned in the Discussion (page 10, lines 417–421, 427–432), when reducing social life causes a decrease in a sense of purpose in life and/or a loss of ikigai, previous researches have shown that not only mental health but also heart disease, stroke, dementia, occurrence of functional impairments, and early death can be related. Therefore, in this sentence in the Abstract, we mentioned broader health. The following revision was made to convey the significance of our study and the characteristics of its results while mentioning mental health. Page 1, lines 25–29 Before revising: “While there is no doubt that infection prevention is important, considering the long-term effects on health, health professionals should help advise older adults wishing to continue activities and maintain a sense of purpose”. After revising: “While there is no doubt that infection prevention is important, supporting older adults in engaging in activities that provide a sense of purpose in life could contribute to their present and future overall health including mental health.”.
|
|
2 |
-Page 3: description of the case (Japan). In the last paragraph of section 1.1., I would be interested to know whether the government targeted, like many countries, a specific group of age. For example, some governments target the 65+ years old, other focus on the 70+, etc. Was there a recurrent group mentioned in the government's communication/ads? |
Thank you for your insightful comment. The Japanese government made the request to all age groups. I have revised the sentence as follows: Page 3, lines 107–108 Before revising: “In Japan, even after the lifting of the state of emergency, the government requested citizens to take the following measures:” After revising: “In Japan, even after the lifting of the state of emergency, the government requested citizens of all ages to take the following measures:”
|
|
3 |
-Table 1 seems to not fit the description of the note. It seems that for every line, the raw N is presented with the % in parentheses. No SD. Also on Table 1: the answer choices for subjective economic status are very odd. the option “Somewhat concerned” is lacking and there is no obvious difference between “a little concerned” and “not so concerned.” Is this a translation issue? It would have been problematic for quantitative purpose but it's not a major issue in that case. |
I am sorry for this careless mistake. Only the average age value shows SD. “Note” in Table 1 has been revised as follows: Page 4, line 139 Before revising: “Note: Data are presented as percentages (%) or mean (SD).” After revising: “Note: Data are presented as percentages (%) or mean [SD].” In addition, regarding subjective economic status, I think it was a translation issue. As per your advice, we have revised “A little concerned” to “Somewhat concerned” and “Not so concerned” to “Minorly concerned” in Table 1, respectively. Table 1 (Page 4, line 138) Before revising: “A little concerned” After revising: “Somewhat concerned” Before revising: “Not so concerned” After revising: “Minorly concerned” |
|
4 |
-Page 6: what do you mean by pensioners? A reader that is not familiar with Japan (like me) would not be sure to understand. Is this similar to long-term health care houses like in the US?
|
Japan also has a long-term health care system, but the pension system is different from it. Based on your advice, I understand that the Japanese pension system needs to be explained. The Japan Pension Service explains the Japanese pension system as follows: “The National Pension is a public pension system participated in by all persons aged 20 to 59 years who have an address in Japan, which provides benefits called the ‘Basic Pension’ due to old age, disability, or death.” The amount of money you receive depends on your subscription period, but for example, if you fully contribute for a maximum of 40 years, you can receive Old-age Basic Pension of 781,700 yen per year starting at age 65. Employees also usually participate in Employees’ Pension Insurance systems. Though the amount of money you can receive differs depending on the period of enrollment and the amount of earned wage, you can receive up to about 3 million yen per year, or on average about 1.2 million yen per year, of Old-age Employees’ Pension. In other words, an average employee can receive an annual Old-age Pension of about 2 million yen starting at age 65. This is the amount of money that generally guarantees the lowest standard of living in Japan. Therefore, several participants noted that the Pension system helped them avoid financial hardship. Reference: Japan pension service. National Pension System. Available online: https://www.nenkin.go.jp/international/japanese-system/nationalpension/nationalpension.html I have added the following explanation regarding the Japanese Pension System: Page 6, line 238 – page 7, line 247 Before revising: “As a background, some participants mentioned that they were pensioners and hence were unlikely to feel the serious changes that would make their lives untenable. Two participants had paid jobs, but none of them were economically disadvantaged by reducing their activities. Therefore, it was said that it did not change in daily life.” After revising: “As a background, some participants mentioned that they were pensioners and hence were unlikely to feel the serious changes that would make their lives untenable. All persons aged 20 to 59 years who have an address in Japan participate in the National Pension, a public pension system, which provides benefits called “Basic Pension” due to old age, disability, or death [30]. In addition, employees also usually enroll in “Employees’ Pension Insurance systems.” Pensioners can receive Old-age Pension each year from age 65, depending on the period of enrollment and the amount of earned wage [30]. The average amount of pension generally guarantees the lowest standard of living in Japan. Two participants had paid jobs, but none of them were economically disadvantaged by reducing their activities. Therefore, it was said that there were no real changes in daily personal life.” |
|
5 |
-I particularly liked the discussion about the male-only comments on page 8 and the relationship between loneliness and living with a parter or not. The distinction between rural/urban was also noteworthy. |
Thank you for your positive feedback. We were also interested in such differences. |
|
6 |
-Page 9, the very end of the first paragraph of section 4. It should be clarified that the impact on mental health (that is mitigated) related to economic security/anxiety. It leaves intact other forms of mental health issues like loneliness (i.e. some can have no economic anxiety but still have mental health issues).
|
According to your advice, I have made the following revision to eliminate ambiguity: Page 9, lines 383–385 Before revising: “Therefore, the impact on mental health might be mitigated compared with the younger generation or those in employment.” After revising: “Therefore, the impact on mental health related to economic security/anxiety might be mitigated compared with the younger generation or those in employment.” |
|
7 |
-About eh fear of “public, watchful eyes”: did they talk about the fear of being judged by their partner? Of course it applies to only those who live with a partner, but I can see a potential fear of being seen as reckless if one ‘goes out’ too often. The fear would not come from the public, but rather within the household.
|
None of participants in this study expressed fear of being judged by their partner or other family members. The following participant’ narrative is quoted in page 7, lines 277–278: “I suddenly lost all schedules, and when I was told not to go out, not to walk, and to stay still at home, it devastated me considerably.” In this quote, it is not clear who said it to the participant, but from the context before and after this quote, it can be understood that it was said by the participant’s family members and close friends. Another participant told his daughter not to use public transportation because he concerned his daughter would be infected. Instead, he drove his daughter to work using his car. Thus, participants in the study showed little fear of being judged by their families, while they seemed to restrict their behavior and that of each other by being worried by and/or worrying about their family members. Based on the above, we interpreted that the “watchful eyes” mentioned by the participants of this study was from the public. |
|
8 |
-The only more important comment that I have is about social desirability bias. The authors do not tackle this issue at all in their discussion about the research design. There are two things. First, I think it should be important to acknowledge that there is a social desirability bias when it comes to mention what kind of activities we have done. See Daoust et al. (2020; in JEPS) for a case study on Canada and/or Daoust et al. (2020; SSRN) for a replication in 12 countries. After all, the authors discuss the potential fear from the public, thus recognizing a social norm. Second, the implications should be discussed. In their case, the fact that the people who were part of the focus group were quite familiar with this exercice, the participants (and maybe the researchers?) might help in the sense that they might feel comfortable to mention freely that they sometimes go out more than they ‘should’.
Again, I believe that this paper is much needed and very important. This is a masterpiece that demonstrates the importance of social sciences in our response to COVID-19. While my comments below would improve the manuscript, I think that they are quite easy to implement (hence the ‘minor revision’ verdict).
|
Thank you for highlighting the very important point of social desirability bias. We read the two papers you shared with great interest. We then reconsidered the results of this study from the viewpoint of social desirability bias. We conducted focus-group interviews using existing groups. Participants were familiar with each other and relatively comfortable discussing issues related to COVID-19 pandemic and public health measures in groups. Therefore, as you mentioned, “they might feel comfortable to mention freely that they sometimes go out more than they ‘should.’” In contrast, it is possible that the “public, watchful eyes” worked also during focus-group interviews among rural participants. In the Discussion, we noted that “public, watchful eyes” made rural participants more sensitive to COVID-19 prevention than urban participants (page 10, lines 441–443). We thought that rural participants might have been more hesitant to talk about their non-compliance behavior due to worry about other participants’ opinions. Based on the above, I added a paragraph regarding social desirability bias and its implications in the Limitation as follows: Page 11, lines 465–481 Before revising: “…that prefer homogeneity and form closed communities. The interviews were conducted 1.5-2.5 months after lifting the state of emergency…” After revising: “…that prefer homogeneity and form closed communities. In the theme of “readiness to endure this restricted life,” participants said that they were prepared to continue to adhere to public health measures even in the long run; however, the impact of the social desirability bias needs to be carefully considered. Social desirability is the pressure for someone to claim to have followed a norm [44]. In this context, Daoust et al. [44] noted the impact of the social desirability bias on the investigation of compliance with COVID-19 public health measures and suggested that people adopt “face-saving” strategies based on their findings. Meanwhile, we conducted focus-group interviews using existing groups. Participants were familiar with each other and relatively comfortable discussing issues related to the pandemic and public health measures in groups. It might have mitigated the impact of the social desirability bias. However, it is possible that the “public, watchful eyes” also worked in this manner during focus-group interviews among rural participants. In other words, the rural participants might have been more resistant toward talking about non-compliant behavior because they were worrying about the “eyes” or opinions of other participants. Thus, the influence of social desirability bias should be considered to a greater extent when interpreting rural participants’ narratives. The interviews were conducted 1.5-2.5 months after lifting the state of emergency…” Again, we are very pleased that you have greatly appreciated our paper. Thank you very much. |
In addition, the entire manuscript was revised to solve grammatical problems and improve readability through detailed English proofreading. These revisions are also colored in red in the main document.
Reviewer 2 Report
In the manuscript “Perception of COVID-19 Restrictions on Daily Life among Japanese Older Adults: A Qualitative Focus Group Study” the authors explore the perceptions of older adults living in urban and rural areas in Japan regarding how their daily lives have been restricted by the spread of COVID-19. The manuscript is well written and deals with a very current and important topic. A minor problem with the work done is that only 8 women (33.3%) were included in the sample and that no systematic comparisons have been made between the responses of women and men. Given the relevance of gender in health, this is a limitation of the study and should be recognized in the section on limitations.
Author Response
|
No. |
Suggestions |
Responses and revisions |
|
1 |
In the manuscript “Perception of COVID-19 Restrictions on Daily Life among Japanese Older Adults: A Qualitative Focus Group Study” the authors explore the perceptions of older adults living in urban and rural areas in Japan regarding how their daily lives have been restricted by the spread of COVID-19. The manuscript is well written and deals with a very current and important topic. A minor problem with the work done is that only 8 women (33.3%) were included in the sample and that no systematic comparisons have been made between the responses of women and men. Given the relevance of gender in health, this is a limitation of the study and should be recognized in the section on limitations.
|
As you advised, we have added that the number of female participants was 33%, which is less than that of male participants, and that gender differences need to be investigated more in the in the Limitations section.
Page 11, lines 487–493 Before revising: “…after lifting the state of emergency was close enough to hear the fresh experiences of older adults. All participants in this study belonged to a community-based group for older adults…” After revising: “…after lifting the state of emergency was close enough to hear the fresh experiences of older adults. Of the 24 participants in the study, only 8 (33%) were women. In the results of this study, the theme of “Concern for a languishing society” was generated only from the narratives of male participants, which shows a gender difference. Further investigation may be necessary from the perspective of gender differences, given the relevance of gender in health and well-being. All participants in this study belonged to a community-based group for older adults…”
|
In addition, the entire manuscript was revised to solve grammatical problems and improve readability through detailed English proofreading. These revisions are also colored in red in the main document.